# A Novel Sugar-Assisted Solvothermal Method for FeF_2_ Nanomaterial and Its Application in LIBs

**DOI:** 10.3390/ma16041437

**Published:** 2023-02-08

**Authors:** Yanli Zhang, Qiang Zhang, Xiangming He, Li Wang, Jingxin Wang, Liangliang Dong, Yingpeng Xie, Yongsheng Hao

**Affiliations:** 1School of Materials Science and Engineering, Shenyang University of Chemical Technology, Shenyang 110142, China; 2Institute of Nuclear and New Energy Technology, Tsinghua University, Beijing 100084, China

**Keywords:** FeF_2_, solvothermal, sugar, sucrose, carbon, lithium-ion battery

## Abstract

Due to its quite high theoretical specific-energy density, FeF_2_ nanomaterial is a good candidate for the cathode material of high-energy lithium-ion batteries. The preparation of FeF_2_ nanomaterial is very important for its application. At present, the preparation process mostly involves high temperature and an inert atmosphere, which need special or expensive devices. It is very important to seek a low-temperature and mild method, without the need for high temperature and inert atmosphere, for the preparation and following application of FeF_2_ nanomaterial. This article reports a novel sugar-assisted solvothermal method in which the FeF_3_∙3H_2_O precursor is reduced into FeF_2_ nanomaterial by carbon derived from the dehydration and condensation of sugar. The obtained FeF_2_ nanomaterials are irregular granules of about 30 nm, with inner pores inside each granule. Electrochemical tests show the FeF_2_ nanomaterial’s potential as a lithium-ion battery cathode material.

## 1. Introduction

The limited reserves of primary energy (coal and oil) and environmental pollution have led to exporting new forms of clean energy (wind, solar, etc.). The intermittency of clean energy requires that the harvested clean energy be stored, mostly in the form of electrical energy. Among all types of energy-storage devices, electrochemical devices are widely used and explored; these devices can store electrical energy in the form of chemical energy and convert chemical energy into electrical energy output [1]. The lithium-ion battery is a typical electrochemical device, which has been commercially used in phones or laptops. Nowadays, clean energy storage and driving electric cars require a higher energy density for lithium-ion batteries [2]. The energy density of a lithium-ion battery is largely determined by the energy density of the electrode material. Current commercial electrode materials are intercalation-type LiCoO_2_ and LiFePO_4_, etc. Electrode materials with higher theoretical energy density, such as sulfur-positive electrodes, are being actively explored. Metal fluorides have high theoretical energy density, attributed to a high capacity (>1 electron per metal ion) derived from a different conversion-type storage mechanism and a high conversion voltage derived from the high iconicity of the metal-fluorine bond [3]. Among all types of metal fluorides, FeF_2_ is a typical conversion electrode material possessing a high energy density of 1519 Wh kg^−1^ [3,4]. In addition, nanocrystallization is very important to shorten the transport path of electrons and lithium ions and thus is beneficial to improve their electrochemical properties as lithium-ion battery cathode material [5,6]. Therefore, FeF_2_ nanomaterial is a good candidate as a high-energy-density cathode for lithium-ion batteries.

Much research on the preparation and performance of FeF_2_ nanomaterials has been carried out, and many results have been achieved. Until now, the synthesis of FeF_2_ nanoparticles could be classified into a one-step method and a two-step method. As shown in Table 1, the one-step method involves FeF_2_ nanoparticles being synthesized directly from a reaction between an Fe-containing precursor and a F-containing precursor, and it mostly occurs in a high-temperature and inert atmosphere and uses complicated electrochemical setups [7,8,9,10,11,12,13,14,15,16]. The two-step method firstly involves the synthesis of a simultaneous Fe- and F-containing precursor and then its conversion into FeF_2_ nanomaterials [17,18]. The conversion usually also occurs in a high-temperature and inert atmosphere. It is necessary to seek a low-temperature and mild conversion method without the need for an inert atmosphere.

In the two-step method, FeF_3_∙3H_2_O is an easily prepared, simultaneous Fe- and F-containing precursor for FeF_2_. FeF_3_∙3H_2_O is converted into FeF_2_ through heat treatment in Ar at 350 °C or in N_2_ at 300 °C. That is, an inert atmosphere and a relatively high temperature are needed, which must be provided by complex sealing setups and an external energy supply. A conversion process with no need for an inert atmosphere and at a relatively lower temperature is pursued to simplify the conversion process and save energy. As is well known, the solvothermal method is widely used in the synthesis of various nanomaterials. This method usually uses simple setups and is conducted at a relatively lower temperature. In addition, the solvothermal method is reported to produce carbon spheres through hydrocarbon sugar [21,22], especially the dehydration and condensation of sucrose, glucose, fructose, etc.. Carbon is reducible at a high temperature [23,24,25]. Based on these factors, it is reasonable to infer that carbon derived from sugar in a solvothermal process may reduce the FeF_3_∙3H_2_O precursor into FeF_2_. This article explores and verifies this conjecture.

This article reports a novel method to synthesize FeF_2_ nanomaterial which involves direct reduction of the precursor FeF_3_∙3H_2_O into FeF_2_ nanomaterial by carbon derived from sugars in a mild solvothermal environment. This method does not involve an inert atmosphere and a high temperature. The concrete conversion mechanism is speculated by combining the characterization results of the intermediate product. FeF_2_ nanomaterial’s application as an electrode material in lithium-ion batteries is primarily explored, and a fairly high initial discharge capacity is shown.

## 2. Experiment

The FeF_3_∙3H_2_O precursor was synthesized through a simple liquid phase coprecipitation method with FeCl_3_∙6H_2_O, HF and CTAB as the iron source, fluorine source and surfactant, respectively [20]. The solvothermal process for the conversion of FeF_3_∙3H_2_O to FeF_2_ was as follows. FeF_3_∙3H_2_O (1.0 g) and sucrose (0.05 g) were added into 45 mL ethanol in a glass beaker. The dispersion was tip-sonicated for 30 min and then transferred into a 50 mL teflon-lined stainless-steel autoclave and sealed. The autoclave was placed into an electric oven and kept at 160 °C for 6h. Besides this temperature, another two solvothermal temperatures, 150 °C and 170 °C, were also tested. After the solvothermal treatment, the produced sample was collected by centrifugation and washed thoroughly with ethanol. The FeF_2_ sample was finally dried at 80 °C for 10h for further characterization. The solvothermal process was also conducted at 120 °C to explore the detailed conversion mechanism.

The crystal structure of the as-prepared sample was characterized by X-ray diffraction (XRD) using D8 ADVANCE (3 KW) with Cu–K radiation in a 2θ range from 10° to 80° at room temperature. The morphology and microstructure were observed with a scanning electron microscope (SEM, Sirion 200) and a transmission electron microscope (TEM, JEOL JEM-2010). The pore- and surface-area analyses were carried out on a Micromeritics instrument (ASAP 2420) with N_2_ as the absorbate at 77 K. The pore distribution and specific surface area were evaluated on the basis of the Barrett–Joyner–Halenda (BJH) desorption pore and the Brunauer–Emmett–Teller (BET) specific surface. The electrochemical performance of the FeF_2_ nanomaterial was evaluated through two-electrode coin-type cells. Specifically, FeF_2_ nanomaterial, acetylene black and poly(tetrafluoroethene) (PTFE) were mixed at a weight ratio of 80:10:10 and were then suppressed and punched into a small disk followed by drying in a vacuum at 120 °C for 24 h. Lithium foil was used as the counter electrode. The electrolyte involves 1 M LiPF6 dissolved in a non-aqueous solvent of ethylene carbonate (EC), dimethyl carbonate (DMC) and ethyl methyl carbonate (EMC) with a volume ratio of 1:1:1. The cells were assembled in an Ar-filled glove box and then galvanostatically charged and discharged using a battery test system (LAND CT2001A model, Wuhan Jinnuo Electronics Co., Ltd., Wuhan, China) at 20 mAh g^−1^ between 1.0 and 4.2 V versus Li/Li^+^.

## 3. Results and Discussion

Figure 1 shows the XRD spectra of the precursor, the intermediate product at 120 °C and the product at 160 °C, which can be indexed as phases FeF_3_∙3H_2_O (JCPDS 32-0464), FeF_3_∙0.33H_2_O (JCPDS 72-1262) and FeF_2_ (JCPDS 45-1062), respectively. XRD results indicate that during the solvothermal process, along with the increase in the solvothermal temperature, the precursor FeF_3_∙3H_2_O firstly loses most of the crystalliferous water and converts to FeF_3_∙0.33H_2_O, which is then successfully reduced to pure and well crystallized FeF_2_ by the carbon derived from sugar under the solvothermal environment above 160 °C. The carbon converted from reducing sugar is not shown in the XRD spectrum, mainly due to the small amount and amorphous state. The solvothermal process was also tested at two other temperatures: a lower temperature of 150 °C and a higher temperature of 170 °C. The phases of the products obtained at 150 °C are a mixture of FeF_3_∙0.33H_2_O and FeF_2_ revealed by XRD, which indicates that FeF_2_ can form at 150 °C, and some FeF_3_∙0.33H_2_O still exists and does not convert into FeF_2_ completely. In contrast, the product obtained at 170 °C is pure FeF_2_ phase. The temperature eventually chosen was 160 °C, because the solvothermal temperature is from an oven heated by electricity, and this lower temperature is enough for the synthesis of pure FeF_2_ and saves electricity at the same time.

Figure 2a–c show the SEM spectra of the precursor FeF_3_∙3H_2_O, the intermediate product FeF_3_∙0.33H_2_O and the FeF_2_ sample. The precursor FeF_3_∙3H_2_O shows short rods with a length of about 1 μm and diameter of about 100–500 nm. Different from the morphology of the precursor, the solvothermal intermediate product—FeF_3_∙0.33H_2_O—comprises granules with irregular shapes and size of about 30 nm. The huge difference in morphology between the precursor FeF_3_∙3H_2_O and the intermediate product FeF_3_∙0.33H_2_O indicates that the solvothermal process for FeF_3_∙3H_2_O is not only a simple dehydration process, but also a morphology-reshaping process. The FeF_2_ sample has almost the same granular morphology as the intermediate product (FeF_3_∙0.33H_2_O), indicating that the reducing reaction with carbon derived from sugar has little effect on the morphology of the intermediate product—FeF_3_∙0.33H_2_O. Figure 2d–f show the TEM, HRTEM and EDS spectra of the obtained FeF_2_ nanomaterials. The TEM image shows the aggregation of FeF_2_ nanoparticles, and the particle size is consistent with that in the SEM image. In addition, there are some points with different contrast inside each grain, so it is inferred that there are holes inside the grain. The HRTEM image shows that there are two kinds of crystal plane spacing, 0.333 nm and 0.271 nm, which correspond to the (110) and (101) crystal planes of FeF_2_, respectively. In addition, there are indeed internal pores about 6 nm inside the FeF_2_ particles. Moreover, the fringe of the particle shows a slightly amorphous state. The EDS spectrum shows C, Cu, Cr, Fe and F elements, of which Cu and Cr come from micro-grid, Fe and F come from FeF_2_ particles, and C comes from the decomposed product of sucrose. Further, EDS reveals that the content of the amorphous carbon is 0.5 wt.%. This content is much lower than that contained in sucrose, which is speculated due to most of the converted substance from sucrose being washed away during the synthesis process. Combined with the HRTEM image, the small amorphous fringe of the particle is deemed to be amorphous carbon adhering to the particles. The N_2_ sorption–desorption isotherms and the pore size distribution of the FeF_2_ nanomaterial are shown in Figure 2g,h. The nitrogen isotherm shows the typical type IV shape, characteristic of materials containing mesopores. The pore-size analysis reveals two pore sizes (5 nm and 20 nm), and the pore volume of 5 nm pores is small, while the pore volume of 20 nm pores is large. The 5 nm pore size is consistent with that shown in the TEM image, confirming that there are indeed 5 nm inner pores inside the FeF_2_ nanoparticles. However, the pore volume of 5 nm pores shown in the pore-size analysis is much smaller than that shown in the TEM image. It is thus concluded that the inner pore wall is not suitable for N_2_ molecules’ permeation during the pore-size analysis, and only a small amount of some thinner pore wall allows N_2_ molecules to permeate; therefore, just a small pore volume is shown in the pore-size analysis. The 20 nm pore size is speculated from the overlapping of FeF_2_ nanoparticles. BET-specific surface area of the FeF_2_ nanomaterial is 44.0 m^2^/g.

The conversion process from the FeF_3_∙3H_2_O short-rod precursor to the granular FeF_2_ nanoparticles with inner pores by the sugar-assisted solvothermal method is illustrated in Figure 3. It is clearly shown that the FeF_3_∙3H_2_O short-rod precursor firstly loses crystalliferous water and converts to FeF_3_∙0.33H_2_O granules. Moreover, it is reported that “FeF_3_∙0.33H_2_O + MWCNTs nanocomposite” can be synthesized with FeF_3_∙3H_2_O + MWCNTs as precursors by a solvothermal process, and the FeF_3_∙0.33H_2_O nanoparticles in “FeF_3_∙0.33H_2_O + MWCNTs nanocomposite” have inner pores due to the loss of crystalliferous water from FeF_3_∙3H_2_O [26]. At the same time, we notice that the inner cavity of FeF_2_ nanoparticles in this work is almost the same to that of FeF_3_∙0.33H_2_O nanoparticles in “FeF_3_∙0.33H_2_O + MWCNTs nanocomposite”. Therefore, it is inferred that in this work, the loss of the crystalliferous water in FeF_3_∙3H_2_O will definitely leave inner pores in the FeF_3_∙0.33H_2_O intermediate granules, besides of the morphology change from short rods to nanogranules. With the further increase in temperature, sucrose is gradually dehydrated, condensed and carbonized. The FeF_3_∙0.33H_2_O granules are reduced to FeF_2_ with the same nanogranular morphology by the carbon derived from the dehydration and condensation of sugar. It should be noted that the very little carbon converted from sucrose is not illustrated due to the small amount. Still, considering that FeF_2_ nanoparticles in this work have inner cavities similar to those of FeF_3_∙0.33H_2_O nanoparticles in “FeF_3_∙0.33H_2_O + MWCNTs nanocomposite”, it is concluded that the reduction effect itself during the FeF_3_∙0.33H_2_O intermediate conversion into FeF_2_ does not change the inner pore in the nanogranules, that is, the inner pores in the FeF_3_∙0.33H_2_O intermediate nanogranules remain in the FeF_2_ product nanogranules. The loss of the small amount of crystalliferous water in FeF_3_∙0.33H_2_O during its conversion into FeF_2_ may induce a slight increase in the inner pores. Above all, the results confirm the successful preparation of FeF_2_ nanomaterials from FeF_3_∙3H_2_O precursor with a sugar-assisted solvothermal method in mild and low-temperature experimental conditions without the use of an inert atmosphere and high temperature. The inner pores in FeF_2_ nanogranules are beneficial to buffer the volume change during the intercalation–deintercalation of lithium and the conversion process when FeF_2_ nanogranules are being used as the cathode in lithium-ion batteries.

The electrochemical properties of the obtained FeF_2_ nanomaterials as a lithium-ion battery cathode material are explored through measuring charge–discharge curves at 20 mA g^−1^ between 1.0 and 4.2 V. One point needing to be explained in advance is that the content of carbon is quite low, so its effect on the electrochemical properties of the FeF_2_ nanomaterials is not considered. As shown in Figure 4a,b, the specific capacity of the first discharge is 561.7 mAh g^−1^, and the first discharge curve is roughly divided into several sections. The first section is a slanted line ranging from 3.0 V to 1.8 V, corresponding to a specific capacity of about 60 mAh g^−1^, attributed to the intercalation of Li^+^ in FeF_2_ to form LiFe_2_F_6_ [27]. The second section is a platform at about 1.8 V and the tailing line until the cut-off voltage, corresponding to a specific capacity of about 500 mAh g^−1^, which is speculated to originate from Fe getting out of the LiFe_2_F_6_ lattice and LiF generation.

In the following cycles, along with the cycling, the discharge capacity of the first section slightly increases and is about 40 mAh g^−1^ at the 50th cycle. Compared with the minor change of the first slanted line section, the specific capacity of the second platform section decreases rapidly and is 10 mAh g^−1^ at the 50th cycle. The corresponding platform potential also decreases rapidly and is about 1.06 V at the 50th cycle. The total discharge-specific capacity at the 50th cycle is 50 mAh g^−1^, and the degradation rate is 1.2% per cycle. The columbic efficiency varies slightly between 90% and 110% from the 2nd cycle. The rate performance of the FeF_2_ nanomaterial is evaluated. From Figure 4c, it can be seen that the nanomaterial exhibits reversible discharge capacities of 440, 240, 90 and 40 mAh g^−1^ at current densities of 28 (0.05C), 60 (0.1C), 140 (0.25C), 285 (0.5C) mA g^−1^, respectively.

The first discharge-specific capacity of the obtained FeF_2_ nanomaterial is close to the theoretical specific capacity of FeF_2_, indicating the feasibility of the obtained FeF_2_ nanomaterial through the sugar-assisted solvothermal method as a lithium-ion battery cathode material. The specific capacity degrades rapidly after the 2nd cycle, and the degradation is similar to that of another pure FeF_2_ nanomaterial synthesized through heat treatment of FeSiF_6_∙6H_2_O in N_2_, whose capacity is 80 mAhg^−1^ after 20 cycles between 1.5 and 4.5 V at 50 mAg^−1^ and at 60 °C [28]. By analyzing the discharge–charge curves during cycling in Figure 4a, the capacity degradation occurs mainly in the second platform, and the rapid degradation of the second platform section indicates that the conversion reaction, that is, LiFe_2_F_6_ → LiF + Fe, is greatly blocked along with cycling. The reason for the blocked conversion reaction here is speculated to be consistent with that in Amatucci’s report, which reveals the CEI on FeF_2_ [28] during cycling under the same electrolyte “LiPF_6_/EC + DMC”: (1) the trapped conversion phases (Fe and LiF) in CEI cannot be involved in the conversion reaction and can no longer contribute to the discharge capacity; (2) LiF in CEI, from the decomposition product of the electrolyte and the trapped conversion phases, has great resistance to Li^+^ transport. That is, the interface reaction between FeF_2_ nanomaterials and the electrolyte “LiPF_6_/EC + DMC” is the main reason leading to the rapid degradation of the capacity and the reduced voltage of the second platform section. There are several strategies reported to hinder the interface reaction effectively: (1) sheathing FeF_2_ nanomaterials with a carbon coating [29]; (2) forming a permeable CEI for Li^+^ around FeF_2_ nanomaterials by using suitable highly concentrated electrolyte “4M LiFSI/DME”, which decomposes in the first several cycles [4]. In the later stage, these strategies are expected to be used to hinder the interface reaction and improve the cycling performance of the FeF_2_ nanomaterial in this research. Considering the FeF_2_ nanomaterial in this research already has the merits of easy preparation (low temperature and no need for an inert atmosphere) and high initial capacity, an improved cycling property will definitely promote its practical application as the cathode in LIBs.

## 4. Conclusions

A novel sugar-assisted solvothermal method is adopted to synthesize FeF_2_ nanomaterial by using FeF_3_∙3H_2_O as a precursor. With the increase in the solvothermal temperature, the FeF_3_∙3H_2_O short-rod precursor firstly loses most of its crystalliferous water and transforms into FeF_3_∙0.33H_2_O nanoparticles, which are then reduced into FeF_2_ nanoparticles by the carbon derived from the dehydration and condensation of sugar above 160 °C. The obtained FeF_2_ nanomaterials are irregular granules of about 30 nm and have inner pores. The method does not involve the inert atmosphere and high temperature that must be provided by complex and expensive devices and is a mild and simple method occurring at a relatively lower temperature. The method may be applied to the preparation of other fluoride nanomaterials. The first discharge-specific capacity of the obtained FeF_2_ nanomaterials is 561.7 mAh g^−1^, close to its theoretical specific capacity, which indicates the great potential of the obtained FeF_2_ nanomaterials as a lithium-ion battery cathode material, although the cycling properties need to be improved. The obtained FeF_2_ nanomaterial prepared by the sugar-assisted solvothermal method here is expected to be used in other new energy fields such as supercapacitors, electrocatalysis, etc.

## Figures and Tables

**Figure 1 materials-16-01437-f001:**
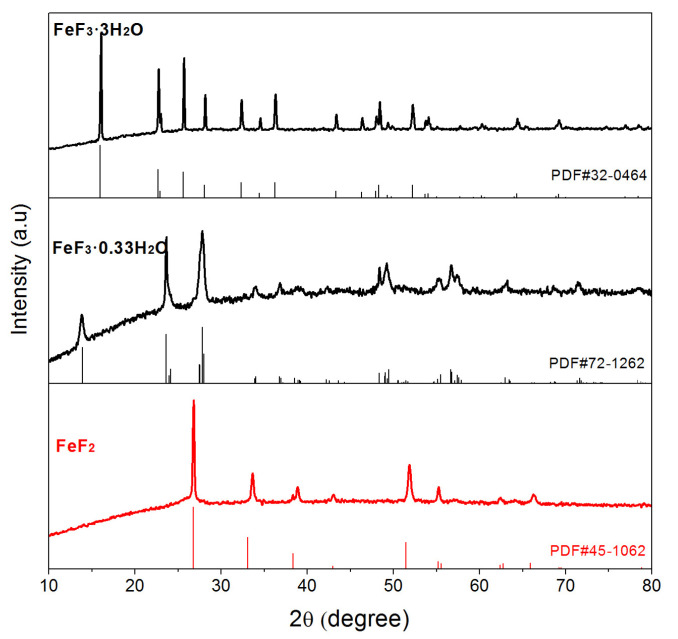
XRD patterns and the standard patterns of the FeF_3_∙3H_2_O precursor, the intermediate product FeF_3_∙0.33H_2_O and FeF_2_ sample.

**Figure 2 materials-16-01437-f002:**
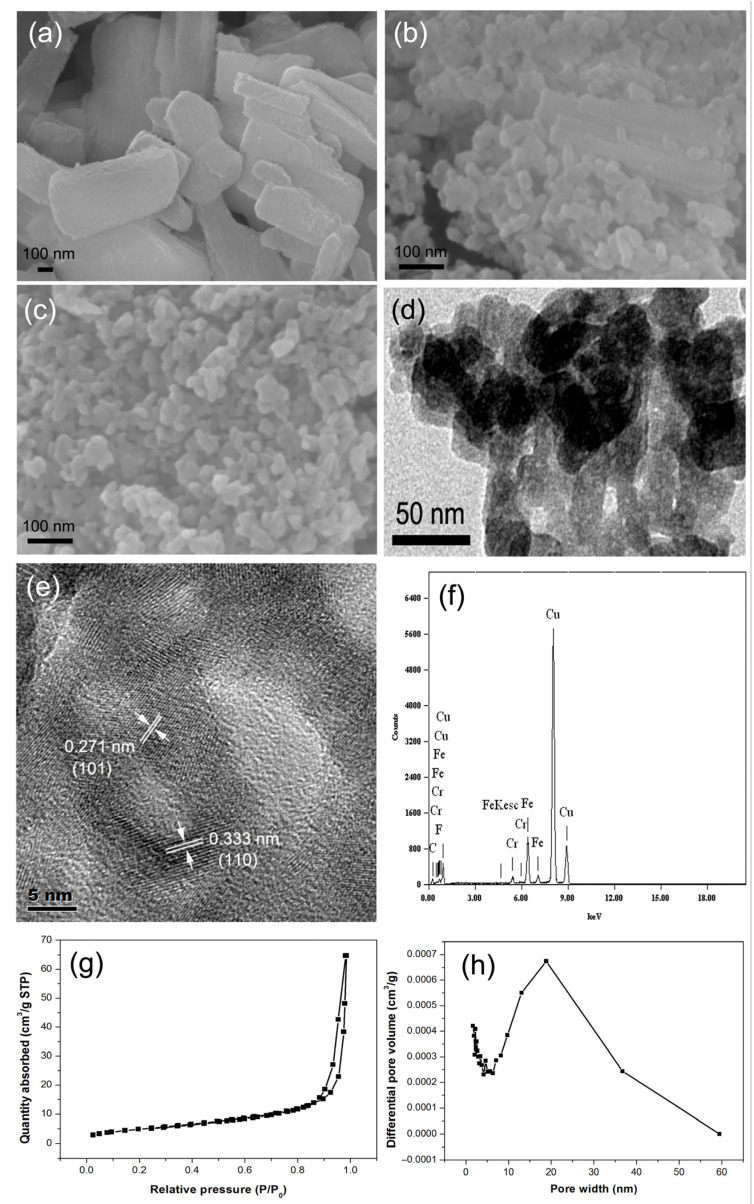
SEM images of the FeF_3_∙3H_2_O precursor (**a**) and the intermediate product FeF_3_∙0.33H_2_O (**b**); SEM image (**c**), TEM image (**d**), HRTEM image (**e**), EDS spectrum (**f**), N_2_ sorption–desorption isotherm (**g**) and the pore size distribution (**h**) of FeF_2_ nanomaterial.

**Figure 3 materials-16-01437-f003:**
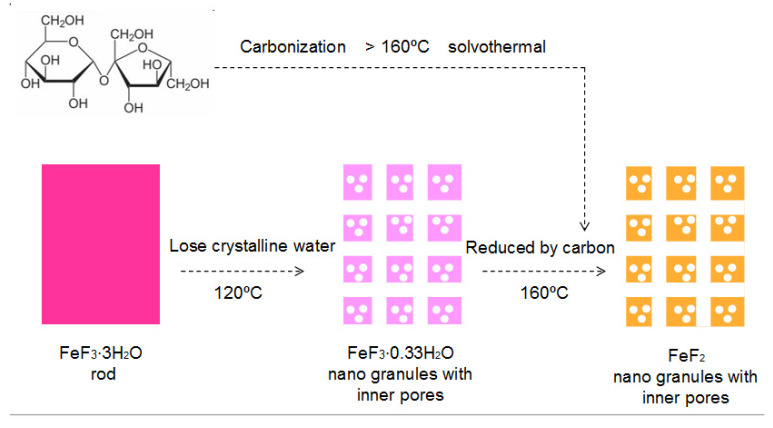
Schematic illustration of the transformation process from the FeF_3_∙3H_2_O precursor, to the intermediate product FeF_3_∙0.33H_2_O, and to FeF_2_ sample.

**Figure 4 materials-16-01437-f004:**
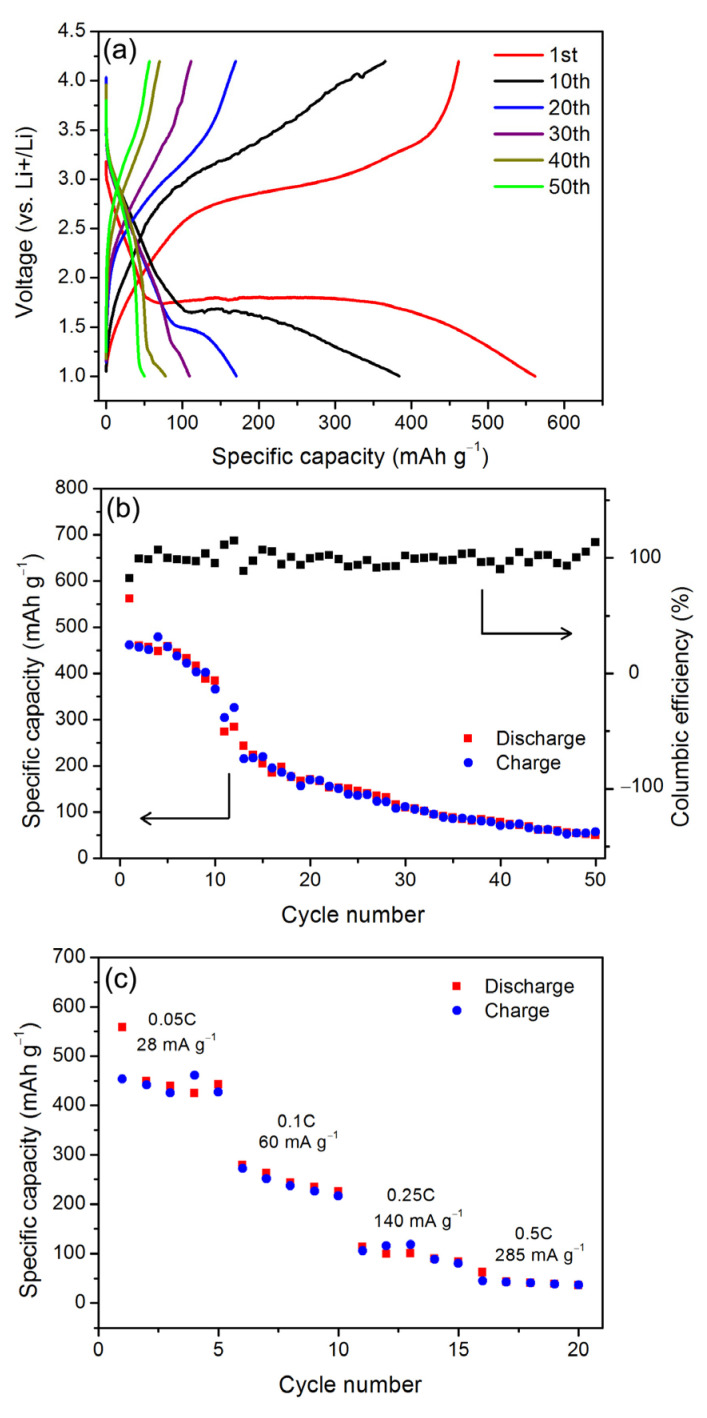
(**a**) 1st, 10th, 20th, 30th, 40th and 50th discharge–charge curves measured at a current rate of 20 mAh g^−1^ between 1.0 V and 4.2 V (vs Li^+^/Li), (**b**) cycling performance and columbic efficiency (black squares), (**c**) rate performance under different current densities of FeF_2_ nanomaterials.

**Table 1 materials-16-01437-t001:** Summary of the existing synthesis methods of FeF_2_ nanomaterials.

One-Step Method						
Precursors			Synthesis Process			Final Product (FeF_2_)
Fe(CO)_5_ + PVDF			Ar 500 °C			FeF_2_@C nanorod [14]
Fe(CO)_5_ + CFx			Ar 250 °C			FeF_2_-C nanocomposite [15]
Fe(ClO_4_)_2_·xH_2_O + CFx			Percolate in CAN, electrochemical			FeF_2_-C nanocomposite [9]
**Two-Step Method**						
**Precursors**			**Synthesis Process**	**Intermediate Phase**	**Synthesis Process**	**Final Product (FeF_2_)**
Fe + H_2_SiF_6_			Liquid synthesis	FeSiF_6_·6H_2_O	Ar 250 °C	FeF_2_ nanoparticle (20 nm) [19]
Fe + H_2_SiF_6_			Liquid synthesis	FeSiF_6_·6H_2_O solution, vacuum infiltration, porous carbon	Ar 250 °C	FeF_2_ in porous carbon [18]
HF + Fe(NO_3_)_3_·9H_2_O			Liquid synthesis	FeF_3_·3H_2_O	Ar 350 °C	FeF_3_ + FeF_2_ (few) [20]
FeCl_3_ solution + CMK-3	hydrothermal	Fe_2_O_3_-CMK-3 + HF	seal	FeF_3_·3H_2_O-CMK-3	N_2_ 300 °C	FeF_2_-CMK-3 [17]

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
