# Peer review of "A Novel Sugar-Assisted Solvothermal Method for FeF2 Nanomaterial and Its Application in LIBs"

_materials, 2023, doi:10.3390/ma16041437_

Round 1
Reviewer 1 Report
The manuscript entitled "A novel sugar-assisted solvothermal method towards FeF2 nanomaterial and its application in LIBs" is devoted to a method for the synthesis of nanosized iron fluoride as a material for the cathode of lithium-ion batteries. The work made a lasting impression on me. I am not ready to recommend the manuscript for publication in its original form. But I am ready to reconsider my decision after the revision of the manuscript.
- How the purity of the electrode powder (presence of sucrose) will be affect the power characteristics of the cell? Compare the cell capacitance values with the literature data for FeF2 cathodes obtained in inert atmospheres.
- Part of the sentence needs to be rewritten because carbon does not decompose "a little of C comes from the decomposed carbon of sucrose"
- In which form the sucrose is present after the synthesis of the main nanopowder? In the form of sucrose crystals, an amorphous powder, or a partially decomposed compound?
- It is necessary to perform a TG study to prove that at the intermediate stage of the synthesis the composition is FeF3∙0.33H2O
- the sentence is incorrect: "the intercalation of Li+ in FeF2 to form LiFeF2 [26]", since in the reaction you proposed, iron lowers the oxidation state from 2+ to 1+, which is unlikely. It is more likely that the reaction of iron disproportionation occurs between 2+, 3+ and 0.
- what are the prospects for the use of your materials, taking into account the strong decrease in capacity after only 30 cycles (Fig. 4)?
Reviewer 2 Report
This manuscript reports on the preparation of FeF2 nanomaterial from direct reduction of FeF3-3H2O precursor with carbon derived from sugar in a mild solvothermal condition and its application to lithium ion battery (LIB) cathode. The obtained FeF2 was irregular granules with size of 30 nm and had inner pores. By applying to LIB cathode, it showed a potential to be utilized as a cathode material. However, the electrochemical property is poor. The following major points should be considered before the acceptance.
1. The used sucrose is dehydrated, condensed, and finally carbonized. How amount of carbon content exists? It is suggested to check the amount of carbon. The amount of the conductive carbon could affect the electrochemical properties.
2. The authors claim the as-prepared FeF2 has inner pore. It is suggested to check the porosity with BET measurement and give the discussion in the manuscript.
3. The reviewer wonders why the authors chose the reaction temperature of 160 degree C. Did the authors try to synthesize the FeF2 materials at other temperatures?
4. The authors synthesized the FeF2 nanomaterials with mild experimental conditions compared to other reported ones. However, the electrochemical performance is very poor. Therefore, the reviewer wonders what the beneficial aspects of the as-prepared materials are for lithium ion batteries. Also, give more detailed discussion on how to improve the electrochemical properties.
Reviewer 3 Report
The writing of this paper is average. The overall quality of your manuscript does not meet the standards of the “Materials” due to the following reasons:
1. The authors should carefully elaborate the advantages of this study compared to the previous papers in literature. Make a comparison table of your work with previous reported literature.
2. Please include the Columbic efficiency variations in discharge results (Fig. 4(a)).
3. Thirty cycles are not enough for publication. At least 50 cycles are required.
4. The most important electrochemical tests such as CV, EIS and rate performances are missing
5. There are several spelling mistakes, please check the English.
Round 2
Reviewer 1 Report
The revised version of the manuscript looks better. I am satisfied with the answers. The manuscript may be considered for publication.
Reviewer 2 Report
The authors have addressed all the questions raised by the reviewer. I recommend the publication of the manuscript in Materials.
Reviewer 3 Report
Accept in present form